# WEBPLANNER: TASK PLANNING WITH AUTONOMOUS EXPERIENCE EXPLORATION AND UTILIZATION FOR REAL WORLD MULTIMODAL WEB AGENTS

## ABSTRACT

Multimodal web agents can assist humans in operating unfamiliar websites and handling repetitive GUI tasks, where effective task planning is essential for decomposing complex tasks into executable actions. While small open-source multimodal large language models (MLLMs) offer a cost-efficient alternative to commercial models, they suffer from weak planning ability and limited generalization especially in cross-website scenarios. To address this, we propose the task decomposition hierarchical analysis framework (TDHAF) to systematically study compositional generalization across three task granularities: low, middle and high levels. And two generalization types: in-domain and out-of-domain. Our analysis reveals that mastering low-level atomic skills does not guarantee high-level planning competence, while high-level task training yields stronger OOD generalization. Motivated by these findings, we introduce the planning experience exploration and utilization (PEEU) method, which enables agents to autonomously set goals, explore unfamiliar environments, and synthesize well-aligned high-level task trajectories from extracted experiences. In real-world multimodal online web navigation, where agents train on one website and are evaluated on 12 unseen websites, PEEU consistently outperforms baselines across model scales (3B, 7B) and training paradigms (SFT, GRPO), reaching 14.9% accuracy, compared to 7.2% and 10.1% for the atomic and basic methods on the GRPO 7B model. These results demonstrate that constructing high-level tasks and leveraging experiences is crucial for OOD planning abilities of small MLLMs.

## 1 INTRODUCTION

The multimodal web agent is an attractive solution, which can assist humans in operating on unfamiliar websites and handling repetitive GUI tasks (Wang et al., 2024; Ning et al., 2025; Tang et al., 2025a). The core ability of the agent is task planning, which enables it to decompose a complex task into executable actions (Li et al., 2025d; Cao et al., 2025; Wei et al., 2025). Due to the high interaction cost of commercial large models, using small open-source multimodal large language models (MLLMs) is a promising approach (Belcak et al., 2025). However, small MLLMs currently exhibit weak planning ability and limited generalization, so it is urgent to enhance their planning abilities (He et al., 2024). In comparison, humans can make plans by utilizing experiences from interaction and exploration with the environment (Ross, 1989; Anderson, 2013). Inspired by the human learning process, agents should (1) set their own learning goals in the environment and improve their abilities through interaction and exploration, and (2) summarize and utilize experiences from the past to guide future decisions (Silver & Sutton, 2025; Cai et al., 2025; Zhang et al., 2025a).

Recent studies focus on utilizing experiences in the post-training stage to further train models. These approaches can be categorized into two main streams: (1) Training with low-level tasks (Gu et al., 2024; Fan et al., 2025). These methods compare changes before and after environment observations to extract experiences. The extracted experiences are then used to synthesize low-level tasks such as clicking, typing, and scrolling to train the model. However, it remains unclear whether training on low-level tasks can effectively generalize to high-level tasks. Hence, it is urgent to propose a framework to study the compositional generalization of web agent task planning. (2) Training with high-level tasks (Logeswaran et al., 2025; Trabucco et al., 2025). These methods leverage task-based

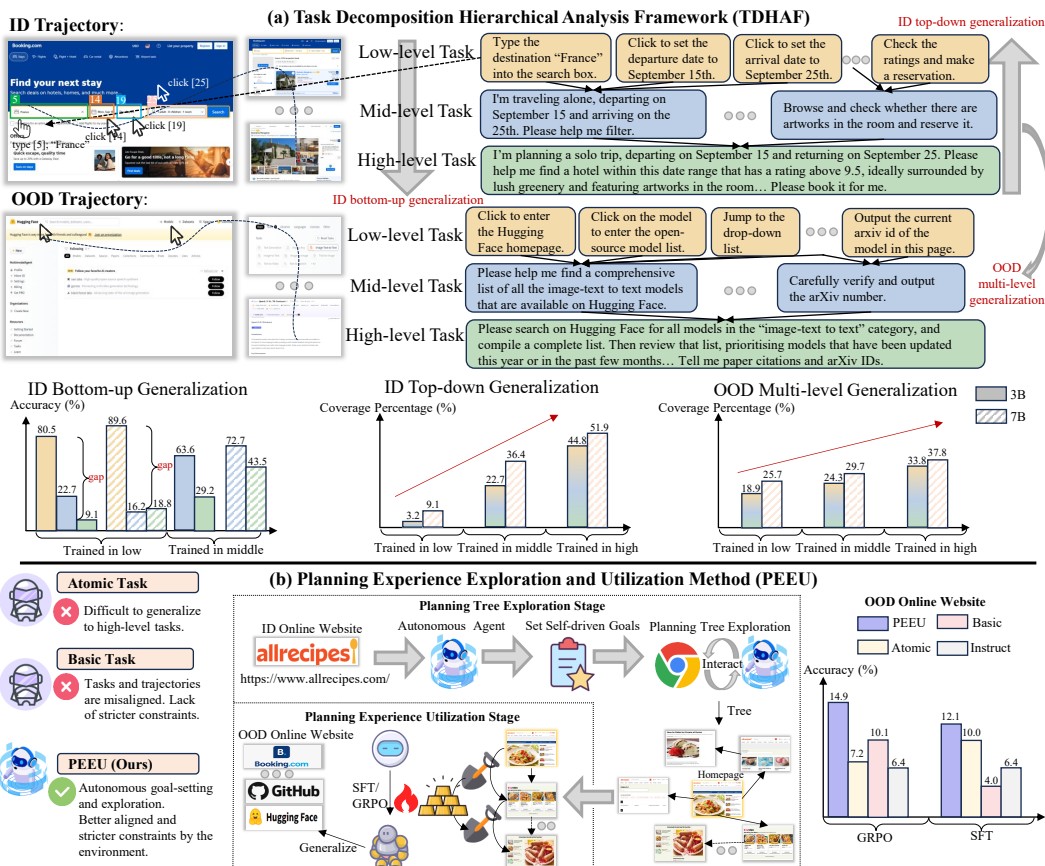

Figure 1: The overview of (a) task decomposition hierarchical analysis framework and (b) planning experience exploration and utilization method.

exploration trajectories to train the model with high-level tasks, like booking a flight with constraints. However, trajectories of high-level tasks suffer from misalignment and a lack of stricter constraints. This limits the generalization ability in high-level tasks. Therefore, it is necessary to develop a method to synthesize trajectories that are better aligned and strictly constrained by environments.

Therefore, we propose the **task decomposition hierarchical analysis framework** (**TDHAF**) to analyze the compositional generalization ability of models in multimodal web navigation planning scenarios, as shown in Figure 1a. This framework first defines three levels of task granularity: **low-level** tasks, **mid-level** tasks, and **high-level** tasks. It further distinguishes between two types of generalization: in-domain (**ID**) and out-of-domain (**OOD**). Building on this taxonomy, we analyze from three perspectives: (1) **ID bottom-up generalization**: whether low-level tasks can generalize to high-level tasks in-domain. (2) **ID top-down generalization**: whether high-level tasks can generalize downwards to low-level tasks in-domain. (3) **OOD multi-level generalization**: what granularity of tasks is better for out-of-domain generalization. The experiments demonstrate the following conclusions: (1) Mastering individual low-level tasks does not necessarily imply mastery of the corresponding high-level task. (2) Using high-level tasks makes it easier to generalize downwards in-domain with greater overall coverage. (3) Using high-level task training can enable the model to acquire stronger generalization capabilities for multi-level tasks in OOD. Overall, experiments show that in the post-training stage, using low-level tasks cannot effectively generalize to high-level tasks.

To enable the agent to have stronger OOD generalization ability, we propose the **planning experience exploration and utilization** method (**PEEU**), as shown in Figure 1b. The framework consists of two stages: planning tree exploration and planning experience utilization. (1) In the **planning tree exploration** stage, the exploration model autonomously sets goals adapted to the functional characteristics of diverse websites, and then conducts goal-driven exploration in the unfamiliar environment to construct an exploration tree. (2) In the **planning experience utilization** stage, trajecto-

ries are summarized to extract valuable experiences. These experiences are then used to create better aligned and constrained pairs of tasks and trajectories. To study the agent's real OOD generalization ability in web navigation, we evaluate it in a multimodal real-world online web setting. The agent trains on one website and tests on 12 completely unseen websites. All methods use the same amount of data and the same hyperparameters to ensure fairness. Based on experiments, our PEEU method explores and utilizes experience automatically, and has stronger cross-website generalization ability. For example, PEEU based on Qwen2.5-VL-7B GRPO reaches 14.9% accuracy, compared to 7.2% for atomic method and 10.1% for basic method. It outperforms baseline methods in both 3B and 7B, SFT and GRPO settings.

In summary, our contributions are as follows: (1) We propose the **task decomposition hierarchical analysis framework** (**TDHAF**) to analyze the compositional generalization ability of models in multimodal web navigation task planning scenarios. (2) We propose the **planning experience exploration and utilization** method (**PEEU**), which can explore and better utilize experiences to improve generalization ability. (3) PEEU improves cross-website OOD generalization in real online multimodal web navigation tasks, outperforming previous methods across different model scales and training settings with the same data scale.

## 2 PRELIMINARIES

In this section, we introduce the definitions of **task planning**, **task levels**, in-domain (**ID**) and out-of-domain (**OOD**). More details and definitions, such as **experience**, are shown in Appendix B.

**Task Planning.** In the ReAct paradigm (Yao et al., 2023), the task planning is defined to decompose a complex task into executable actions, which can be formalized as:

$$a_t = \pi(d, \mathcal{H}_{0:t}, s_t), \tag{1}$$

where $d$ is the task description, $\mathcal{H}_{0:t} = \{(s_0, a_0), \ldots, (s_{t-1}, a_{t-1})\}$ is the history, $s_t$ is the current observation, and $\pi$ is the policy. The complete trajectory is $\tau = \{(s_0, a_0), \ldots, (s_m, a_m)\}$.

**Task Levels.** We define three levels of tasks as shown in Figure 1a. **Low-level Task**: an atomic task at step $t$ uses only the low-level description and current observation, expressed as $a_t = \pi(d_{low}, s_t)$. **Mid-level Task**: a multi-step subtask is executed with the mid-level description, history from $p$ to $t$, and current observation, expressed as $a_t = \pi(d_{mid}, \mathcal{H}_{p:t}, s_t)$. **High-level Task**: a long-horizon task is executed with the high-level description, full history, and current observation, expressed as $a_t = \pi(d_{high}, \mathcal{H}_{0:t}, s_t)$.

**ID and OOD.** ID evaluation uses test data from the same trajectories or websites seen during training, while OOD evaluation uses test data from entirely new and different websites not encountered during training. More details are shown in Appendix B.

## 3 TASK DECOMPOSITION HIERARCHICAL ANALYSIS FRAMEWORK

To analyze the hierarchical generalization capabilities of task decomposition, we propose the **task decomposition hierarchical analysis framework** (**TDHAF**), as shown in Figure 2. This framework provides an analysis from three perspectives: **ID bottom-up generalization**, **ID top-down generalization**, and **OOD multi-level generalization**. The subsequent sections will introduce the analysis framework, data construction, experimental settings, results and analysis.

### 3.1 ANALYSIS FRAMEWORK

To investigate the compositional generalization ability of models in multimodal web navigation task planning scenarios, we propose the task decomposition hierarchical analysis framework. This framework first defines three levels of task granularity: **low-level** tasks, **mid-level** tasks, and **high-level** tasks. It further distinguishes between two types of generalization: in-domain (**ID**) and out-of-domain (**OOD**). Building on this taxonomy, the framework analyzes from three perspectives: bottom-up generalization in-domain, top-down generalization in-domain, and multi-level generalization out-of-domain. Figure 2 provides a detailed example of the analysis framework. Table 3 illustrates the training and testing set divisions for the three generalization dimensions. Further explanations of the three dimensions of generalization are presented following.

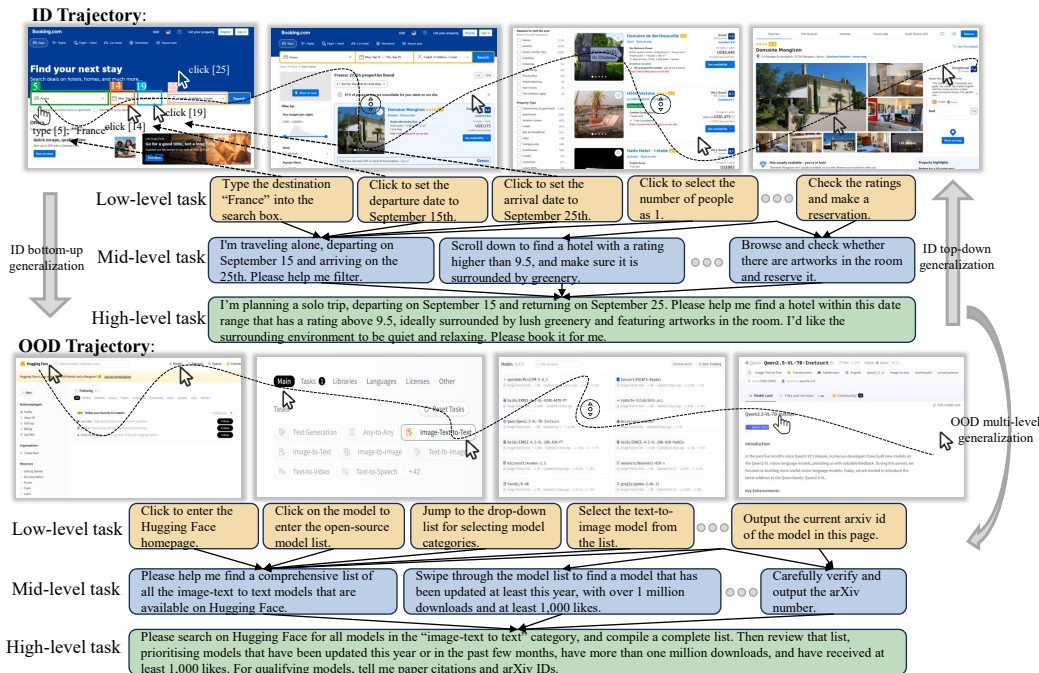

Figure 2: This figure illustrates the task decomposition hierarchical analysis framework. The upper part shows the trajectory of ID, and the lower part shows the trajectory of OOD. Both domains contain three levels: low, middle, and high. We study three generalization dimensions, including ID bottom-up generalization, ID top-down generalization and OOD multi-level generalization.

**ID Bottom-up Generalization.** To study whether the model can generalize from low-level tasks to higher-level composite tasks in-domain, we use relatively low-level tasks as the training set and relatively high-level tasks as the test set. For example, after the model learns single-step atomic task mapping, we test if it can generalize to multi-step subtasks and long-horizon task decomposition. We also test if it can generalize to long-horizon task decomposition after learning subtasks.

**ID Top-down Generalization.** To study whether the model can generalize from high-level tasks to lower-level tasks in-domain, we use relatively high-level tasks as the training set and relatively low-level tasks as the test set, which is the opposite of the previous experiment. For example, after the model learns to decompose long-horizon tasks, we check whether it truly learns the corresponding subtasks and atomic skills.

**OOD Multi-level Generalization.** To study whether the model can generalize task decomposition ability from in-domain tasks to out-of-domain tasks, we separately use three levels of in-domain tasks as the training set. We use unseen cross-website tasks as the test set to evaluate multi-level out-of-domain generalization. For example, after the model learns task decomposition at different levels, we examine how well it applies this ability to unseen tasks.

### 3.2 DATA CONSTRUCTION

Raw data is collected from Multimodal-Mind2Web (Deng et al., 2023; Zheng et al., 2024a). It is an offline human-expert-annotated gold trajectory dataset. Employing such a dataset for analysis offers more significant advantages, as it enables fine-grained examination of the model's behavior at the single-step level, including the target numbers, action types, action parameters. The in-domain test and train data come from the same trajectory, while the out-of-domain test data come from different trajectories of completely different websites. The in-domain training and test data are derived from the same trajectories, but the questions are rewritten. The training set has 616 samples, and the test set has 684 samples. The data statistics are shown in Figure 6. The data split is shown in Table 3. The prompts for generating data are shown in Appendix D, which are the prompts for generating low-level tasks and high-level tasks by GPT-4o.

Table 1: Accuracy comparison across different generalization dimensions. 3B Instruct refers to the Qwen2.5-VL-3B-Instruct model. 3B Low refers to the Qwen2.5-VL-3B-Instruct trained at the low level. Test-ID-Low denotes the in-domain low-level test set. Test-OOD-Low denotes the out-of-domain low-level test set. The bolded entries indicate the model that achieves the highest Step SR among the four models on each test set under the same base model.

| Model | Test-ID-Low | | | | Test-ID-Middle | | | | Test-ID-High | | | |
|---|---|---|---|---|---|---|---|---|---|---|---|---|
| | Id | Action | Value | **Step SR** | Id | Action | Value | **Step SR** | Id | Action | Value | **Step SR** |
| 3B Instruct | 30.3 | 39.5 | 85.7 | 17.8 | 17.1 | 6.6 | 9.5 | 0.0 | 14.4 | 9.6 | 6.7 | 0.7 |
| 3B Low | 81.2 | 99.4 | 100.0 | **80.5** | 28.6 | 83.1 | 4.3 | 22.7 | 12.3 | 85.1 | 0.0 | 9.1 |
| 3B Middle | 72.7 | 98.7 | 95.7 | 71.4 | 66.9 | 95.5 | 73.9 | **63.6** | 32.5 | 85.1 | 0.0 | 29.2 |
| 3B High | 77.3 | 98.1 | 95.7 | 75.3 | 57.8 | 94.2 | 65.2 | 54.5 | 64.9 | 95.5 | 65.2 | **63.0** |
| 7B Instruct | 59.1 | 84.4 | 73.9 | 49.4 | 43.1 | 41.2 | 27.3 | 17.6 | 35.8 | 44.4 | 20.0 | 13.2 |
| 7B Low | 90.3 | 99.4 | 100.0 | **89.6** | 37.7 | 39.6 | 43.5 | 16.2 | 29.2 | 75.3 | 13.0 | 18.8 |
| 7B Middle | 87.0 | 99.4 | 95.7 | 86.4 | 78.6 | 92.2 | 65.2 | **72.7** | 46.1 | 89.6 | 21.7 | 43.5 |
| 7B High | 85.1 | 98.1 | 87.0 | 83.1 | 69.5 | 89.6 | 39.1 | 63.6 | 76.6 | 92.2 | 56.5 | **72.1** |

| Model | Test-OOD-Low | | | | Test-OOD-Middle | | | | Test-OOD-High | | | |
|---|---|---|---|---|---|---|---|---|---|---|---|---|
| | Id | Action | Value | **Step SR** | Id | Action | Value | **Step SR** | Id | Action | Value | **Step SR** |
| 3B Instruct | 40.5 | 63.5 | 100.0 | 31.1 | 21.9 | 20.5 | 33.3 | 6.8 | 16.4 | 16.4 | 22.2 | 0.0 |
| 3B Low | 81.1 | 98.6 | 100.0 | 79.7 | 37.8 | 75.7 | 12.5 | 35.1 | 29.7 | 78.4 | 0.0 | 25.7 |
| 3B Middle | 70.3 | 100.0 | 100.0 | 70.3 | 48.6 | 79.7 | 12.5 | **44.6** | 32.4 | 78.4 | 0.0 | 31.1 |
| 3B High | 82.4 | 100.0 | 100.0 | **82.4** | 45.9 | 81.1 | 12.5 | **44.6** | 39.2 | 81.1 | 6.2 | **39.2** |
| 7B Instruct | 63.5 | 91.9 | 62.5 | 56.8 | 46.6 | 72.6 | 20.0 | 30.1 | 30.1 | 64.4 | 20.0 | 16.4 |
| 7B Low | 89.2 | 97.3 | 93.8 | **85.1** | 56.8 | 78.4 | 31.2 | 50.0 | 37.8 | 79.7 | 18.8 | 33.8 |
| 7B Middle | 83.8 | 100.0 | 93.8 | 82.4 | 59.5 | 82.4 | 12.5 | 51.4 | 37.8 | 78.4 | 0.0 | 35.1 |
| 7B High | 81.1 | 95.9 | 75.0 | 77.0 | 58.1 | 82.4 | 12.5 | **54.1** | 45.9 | 81.1 | 6.2 | **43.2** |

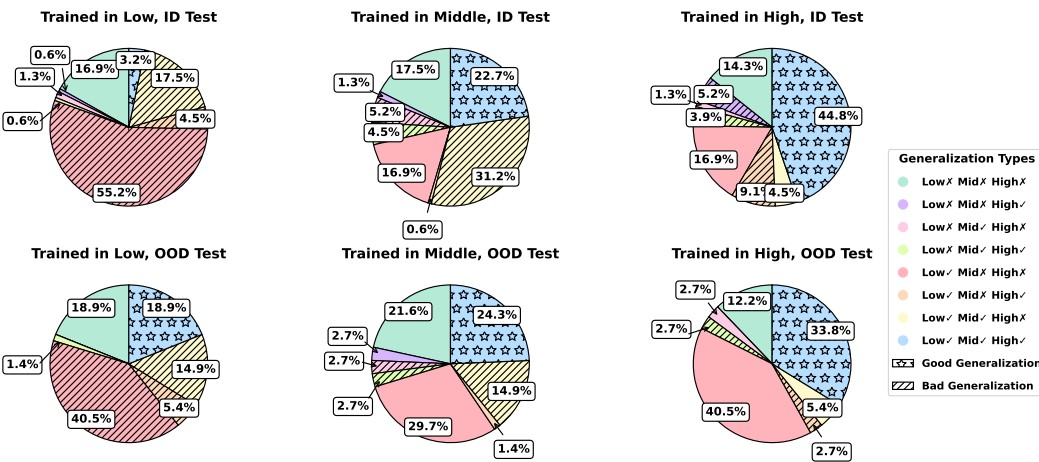

Figure 3: Generalization distribution pie chart for Qwen2.5-VL-3B. The table shows the distribution of eight types of generalization. Good generalization means successful generalization to other levels, the larger the better. Bad generalization means failure to fully generalize, the smaller the better. Results for Qwen2.5-VL-7B, the definitions of good/bad generalization are shown in Appendix E.

## 3.3 EXPERIMENTAL SETTINGS

**Settings.** All experiments are conducted on Qwen2.5-VL-3B-Instruct and Qwen2.5-VL-7B-Instruct for SFT. The batch size is 8, the learning rate is 5.0e-6 and the training epochs are 3, with llama-factory (Zheng et al., 2024b) framework. All experiments are conducted on 4 A800 GPUs.

**Metric.** Following (Deng et al., 2023; Zheng et al., 2024a), we calculate the accuracy between predictions and ground truth, which includes the following four sub-metrics: *Id* refers to the accuracy of interactive element number in the Set-of-Mark (SoM). *Action* measures the accuracy of action types. *Value* evaluates the accuracy of action parameters. *Step SR* represents the accuracy rate of a single-step prediction completely matching the ground truth.

## 3.4 Results and Analysis

**Mastering individual low-level tasks does not necessarily imply mastery of the corresponding high-level task.** As shown in Table 1 and Figure 1a in the Step SR in-domain setting, the 3B-model trained in low-level training data achieves 80.5% accuracy in low-level test tasks, but only 9.1% accuracy for the corresponding high-level test tasks. The 7B-model trained in low-level training data achieves 89.6% accuracy in low-level test tasks, but only 18.8% accuracy for the corresponding high-level test tasks. This shows that the bottom-up post-training method is not an effective way for enhancing planning ability.

**Using high-level tasks makes it easier to generalize downwards in-domain with greater overall coverage.** As shown in Figure 3 and Figure 7 in the in-domain setting, we define a task where all levels succeed as good generalization, and we refer to this percentage as the coverage percentage (Appendix E for a formal definition). For the 3B model, the coverage percentage is 44.8% when trained on high-level tasks, 22.7% on middle-level, and 3.2% on low-level tasks. For the 7B model, the coverage percentage is 51.9% when trained on high-level tasks, 36.4% on middle-level, and 9.1% on low-level tasks. This shows top-down generalization has higher coverage percentage in-domain.

**Using high-level task training can enable the model to acquire stronger generalization capabilities for multi-level tasks in OOD.** As shown in Figure 3 and Figure 7 in the out-of-domain setting, for the 3B model, the coverage percentage is 33.8% when trained on high-level tasks, 24.3% on middle-level, and 18.29% on low-level tasks. For the 7B model, the coverage percentage is 37.8% when trained on high-level tasks, 29.7% on middle-level, and 25.7% on low-level tasks. This shows that top-down generalization also has higher coverage percentage out-of-domain.

## 4 Planning Experience Exploration and Utilization

In this section, we introduce the planning experience exploration and utilization method. This is an automatic exploration learning framework that first sets goals adaptively and explores in unfamiliar websites. Then it extracts planning experiences from trajectories and uses them to build aligned and constrained training data. Users only need to provide a URL to be explored, and the framework can freely explore the website, extract and summarize experiences, and then build better aligned and constrained data to train small MLLMs, achieving cross-website generalization capabilities.

### 4.1 Method

The framework is divided into two stages: **planning tree exploration** and **planning experience utilization**, as shown in Figure 4. All prompts are shown in Appendix F.

**Planning Tree Exploration.** The autonomous agent requires a shift from passive learning to autonomous learning. It requires self-driven tasks and self-execution exploration. For the self-driven tasks stage, given a website URL, the exploration agent interacts with the homepage $s_0$ (obtained from the URL) through the MLLM $M$ to generate a basic task list $\mathcal{D} = \{d_1, d_2, \ldots, d_n\}$, where each task $d_i$ represents a task to be explored. This process can be expressed as:

$$\mathcal{D} = M(s_0, \text{URL}). \tag{2}$$

Subsequently, for the self-execution exploration stage, the agent performs autonomous exploration based on the task list $\mathcal{D}$, the environment Env (with basic URL as entry point), generating a directed exploration tree $\mathcal{R} = (V, E)$ rooted at the homepage, where $V$ is the set of website screens, $E$ is the set of actions between these observations. The exploration process is implemented as:

$$\mathcal{R} = \text{Explore}(M, \mathcal{D}, \text{Env}, \text{URL}). \tag{3}$$

This tree can be expanded into interleaved trajectories of observations and actions, where all trajectories share the same root node. Formally, let $\tau = \{(s_0, a_0), \ldots, (s_m, a_m)\}$ denote a trajectory, where $s_0$ is the shared root state (homepage). $a_t \in \mathcal{A}$ represents the action at step $t$. $s_{t+1} \sim P(\cdot | s_t, a_t)$ is the subsequent observation. The exploration tree $\mathcal{R}$ represents the collection of trajectories from tasks $\{\tau_i\}_{i=1}^n$, obtained via the recursive exploration process by $M$.

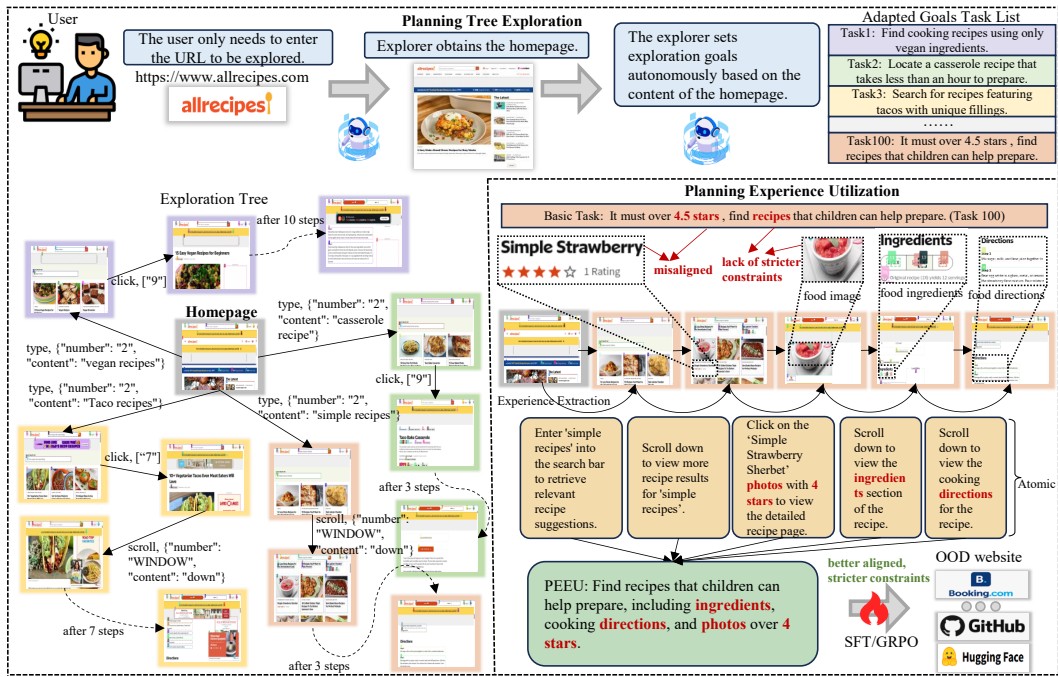

Figure 4: An overview of planning experience exploration and utilization method.

**Planning Experience Utilization.** The agent needs to learn from past explorations and use these experiences to build high-level trajectory data. The basic high-level tasks have two limitations. (1) The tasks and the trajectories are not always aligned. For example, the task requires more than 4.5 stars, but the trajectory only reaches 4 stars. (2) The task lacks stricter constraints for unknown environments, because the websites are naturally partially observable environments. The constraints of the unknown environment must come from real exploration, and the homepage information cannot provide them, such as food ingredients and preparation directions.

In the experience extraction stage, the MLLM $M$ compares before-action state and after-action state to extract atomic experiences:

$$\epsilon_t = M(s_t, a_t, s_{t+1}), \tag{4}$$

where $s_t$ and $s_{t+1}$ are the visual observations before and after action $a_t$, respectively. A trajectory-level experience $\mu$ can be represented as a sequence of atomic experiences:

$$\mu = (\epsilon_1, \epsilon_2, \ldots, \epsilon_T). \tag{5}$$

The agent then fuses these sequences of atomic experiences into refined high-level tasks that are both more aligned with real outcomes and stricter in the constraints. Formally, define a mapping $\Phi$ with $M$ that aggregates the experiences into PEEU task $\tilde{d}$, forming the collection $\tilde{\mathcal{D}}$ of PEEU tasks:

$$\tilde{\mathcal{D}} = (\tilde{d}_1, \tilde{d}_2, \ldots, \tilde{d}_n) = \Phi(\mu_1, \mu_2, \ldots, \mu_n, M). \tag{6}$$

In the training stage, the agent's goal is to learn a policy $\pi : \mathcal{S} \times \mathcal{H} \times \tilde{\mathcal{D}} \to \mathcal{A}$, that maps the current state $s_t \in \mathcal{S}$, the history $h_t \in \mathcal{H}_{0:t}$, and the task description $\tilde{d} \in \tilde{\mathcal{D}}$, to the next action $a_t \in \mathcal{A}$. We use SFT and GRPO (Shao et al., 2024) for training. The details are shown in Appendix F and G.

### 4.2 EXPERIMENTAL SETTINGS

**Baseline.** (1) Atomic-Prompt uses the input task to retrieve related atomic experiences. The retriever uses all-roberta-large-v1 (Reimers & Gurevych, 2020). The number of retrieved atomic experiences is set to 10. These experiences are used as prompts to serve as contextual input. (2) Trajectory-Prompt uses the input task to retrieve one trajectory-level experience according to its query as the prompt. (3) Basic uses the original exploration task as the training task. (4) Atomic uses the atomic operation task as the training task. In addition, all the training parameters are kept the same. And all methods are controlled to use the same amount of data to ensure a fair comparison.

| Model | Method | Rec ID | Ama OOD | App OOD | ArX OOD | Git OOD | Boo OOD | ESP OOD | Cou OOD | BBC OOD | Fli OOD | Map OOD | Hug OOD | Wol OOD | Overall Total |
|---|---|---|---|---|---|---|---|---|---|---|---|---|---|---|---|
| GPT-4o | Instruct | 56.3 | 53.7 | 56.6 | 60.5 | 57.7 | 43.9 | 44.0 | 65.1 | 54.8 | 28.6 | 56.9 | 42.6 | 65.2 | 52.7 |
| Claude 3 Opus | Instruct | 45.9 | 58.6 | 58.1 | 55.0 | 56.9 | 19.0 | 46.2 | 68.2 | 66.7 | 15.1 | 55.3 | 53.5 | 51.5 | 50.0 |
| | Instruct | 0.0 | 0.0 | 0.0 | 0.0 | 0.0 | 0.0 | 0.0 | 0.0 | 2.3 | 0.0 | 0.0 | 0.0 | 2.1 | 0.3 |
| | Atomic-Prompt | 0.0 | 0.0 | 0.0 | 0.0 | 0.0 | 0.0 | 0.0 | 0.0 | 0.0 | 0.0 | 0.0 | 0.0 | 0.0 | 0.0 |
| | Trajectory-Prompt | 0.0 | 0.0 | 0.0 | 0.0 | 0.0 | 0.0 | 0.0 | 0.0 | 0.0 | 0.0 | 0.0 | 0.0 | 0.0 | 0.0 |
| | Basic-SFT | 0.0 | 0.0 | 0.0 | 2.3 | 2.4 | 0.0 | 0.0 | 4.7 | 0.0 | 0.0 | 2.4 | 6.9 | 4.3 | 1.7 |
| Qwen2.5-VL-3B | Basic-GRPO | 0.0 | 2.4 | 0.0 | 20.9 | 0.0 | 2.2 | 0.0 | 2.3 | 0.0 | 2.3 | 2.4 | 0.0 | 17.3 | 3.8 |
| | Atomic-SFT | 2.2 | 2.4 | 0.0 | 4.6 | 7.3 | 2.2 | 2.2 | 7.1 | 2.3 | 2.3 | 0.0 | 2.3 | 15.2 | 3.8 |
| | Atomic-GRPO | 0.0 | 12.1 | 2.3 | 11.6 | 0.0 | 2.2 | 0.0 | 9.5 | 0.0 | 4.7 | 0.0 | 6.9 | 8.6 | 4.4 |
| | PEEU-SFT (Ours) | 2.2 | 7.3 | 6.9 | 11.6 | 2.4 | 12.9 | 4.5 | 0.0 | 7.1 | 2.3 | 4.8 | 2.3 | 10.8 | _5.7_ |
| | PEEU-GRPO (Ours) | 6.6 | 24.3 | 3.0 | 23.2 | 9.7 | 6.8 | 2.2 | 7.1 | 0.0 | 2.3 | 0.0 | 0.0 | 15.2 | **7.7** |
| | Instruct | 2.2 | 7.3 | 9.3 | 4.6 | 9.7 | 0.0 | 0.0 | 16.6 | 7.1 | 0.0 | 0.0 | 13.9 | 13.0 | 6.4 |
| | Atomic-Prompt | 2.2 | 0.0 | 6.9 | 4.6 | 2.4 | 0.0 | 0.0 | 9.5 | 0.0 | 0.0 | 0.0 | 0.0 | 4.3 | 2.3 |
| | Trajectory-Prompt | 4.4 | 0.0 | 0.0 | 4.6 | 2.4 | 0.0 | 0.0 | 9.5 | 0.0 | 2.3 | 2.4 | 0.0 | 6.5 | 2.4 |
| | Basic-SFT | 0.0 | 4.8 | 0.0 | 4.6 | 0.0 | 0.0 | 0.0 | 7.1 | 0.0 | 0.0 | 4.8 | 13.9 | 17.3 | 4.0 |
| Qwen2.5-VL-7B | Basic-GRPO | 0.0 | 17.0 | 7.1 | 20.9 | 4.8 | 13.6 | 0.0 | 4.7 | 4.7 | 2.3 | 12.1 | 18.6 | 26.0 | 10.1 |
| | Atomic-SFT | 15.5 | 17.0 | 11.6 | 23.2 | 0.0 | 4.5 | 0.0 | 7.1 | 0.0 | 4.7 | 4.8 | 23.2 | 19.5 | 10.0 |
| | Atomic-GRPO | 2.2 | 19.5 | 0.0 | 18.6 | 0.0 | 9.0 | 2.2 | 11.9 | 0.0 | 2.3 | 0.0 | 0.0 | 28.2 | 7.2 |
| | PEEU-SFT (Ours) | 8.8 | 24.3 | 18.6 | 16.2 | 7.3 | 2.2 | 2.2 | 16.6 | 14.2 | 2.3 | 7.3 | 11.6 | 26.0 | _12.1_ |
| | PEEU-GRPO (Ours) | 4.4 | 26.8 | 18.6 | 20.9 | 21.9 | 6.8 | 0.0 | 33.3 | 26.1 | 0.0 | 12.1 | 2.3 | 21.7 | **14.9** |

Table 2: Performance across different websites. Bold indicates the highest performance. Underline indicates the second-highest performance. Overall is the average accuracy of all websites.

**Evaluation.** We evaluate the planning capabilities of the models on real-world multimodal benchmark WebVoyager (He et al., 2024). The test set covers diverse real multimodal online websites, including cooking, shopping, research, code, travel, sports, news, map, study and other categories. Follow the standard evaluation procedure of WebVoyager (He et al., 2024), the benchmark uses the trajectory-level success rate as the final accuracy.

**Exploration and training settings.** (1) For the exploration phase, we use GPT-4o for exploration with a maximum step length of 15 in 100 exploration tasks. The browser observation resolution is set to 1024*768 pixels. For the experience summarization phase, we use GPT-4o to summarize the changes in the browser's state before and after the exploration. (2) For the training phase, all our experiments are conducted on Qwen2.5-VL-3B-Instruct and Qwen2.5-VL-7B-Instruct. For the SFT model, the batch size is 16, the learning rate is 5.0e-6, and the number of training epochs is 5, using the llama-factory (Zheng et al., 2024b) training framework. For the GRPO model, the batch size is 20, the learning rate is 1.0e-6, the rollout size is 10, and the number of training epochs is 7, using the verl (Yaowei Zheng, 2025) framework. All experiments are performed on 4 A800 GPUs. For fair comparison, all experiments use identical trajectories. (3) For the division of training and testing, to thoroughly validate the model's generalization and universal capabilities, we trained exclusively on the Allrecipes (Rec) website, while the remaining 12 websites were unseen during the training to test ability to generalize OOD. More details are shown in Appendix H.

## 4.3 RESULTS AND ANALYSIS

**Adapt the task to fit the trajectory with experience.** As shown in Figure 4, basic trajectory tasks face problems of mismatch and a lack of strict constraints. For example, in the basic task, the rating is 4.5, but the trajectory shows only 4 stars, which causes a mismatch. In addition, the basic task lacks exploration of the model environment, so it lacks environmental constraints. Therefore, constraints should be derived from exploration experience. By using experience to modify tasks, we can create more aligned and strictly constrained advanced tasks. As shown in Table 2, for the 3B model, the SFT and GRPO of our PEEU are 5.7% and 7.7%, higher than 1.7% and 3.8% of the basic task. For the 7B model, the SFT and GRPO of our PEEU are 12.1% and 14.9%, higher than 4.0% and 10.1% of the basic task. This proves the effectiveness of adapting tasks with experience.

**Using higher-level tasks provides better cross-website generalization than lower-level tasks in real-world websites.** As shown in Table 2, we train only on the Rec website and test on 12 other websites that the model never sees in training stages to fully test cross-website generalization. For the 3B model, our PEEU reaches 5.7% in SFT and 7.7% in GRPO, higher than the low-level scores of 3.8% and 4.4%. For the 7B model, our PEEU reaches 12.1% in SFT and 14.9% in GRPO, higher than the low-level scores of 10.0% and 7.2%. This shows that using more aligned and constrained trajectories makes models stronger in cross-website generalization than in low-level tasks.

**Without a specially designed prompt pipeline, direct training is more effective than retrieval for small models.** As shown in Table 2, we apply both training and retrieval under the same experiences. Because of the limited ability of small models, using prompts without changing model parameters does not effectively help them improve in complex tasks. For example, with the retrieval method, a 7B model gets scores of 2.3% and 2.4%, which are even lower than the base model score of 6.4%. This shows direct training is more effective than retrieval for small models.

**PEEU has the capability for more effective long-horizon planning.** As shown in Figure 5 and Figure 8, the x-axis denotes the steps of successful trajectories, while the y-axis denotes the count of successful trajectories. PEEU enables more effective long-horizon planning, thanks to the higher-quality high-level data. In contrast, atomic approaches constrain the model's ability to generalize over long-horizon planning, with most being limited to completing only 2-step tasks. These results show that PEEU demonstrates a much stronger advantage in long-horizon planning.

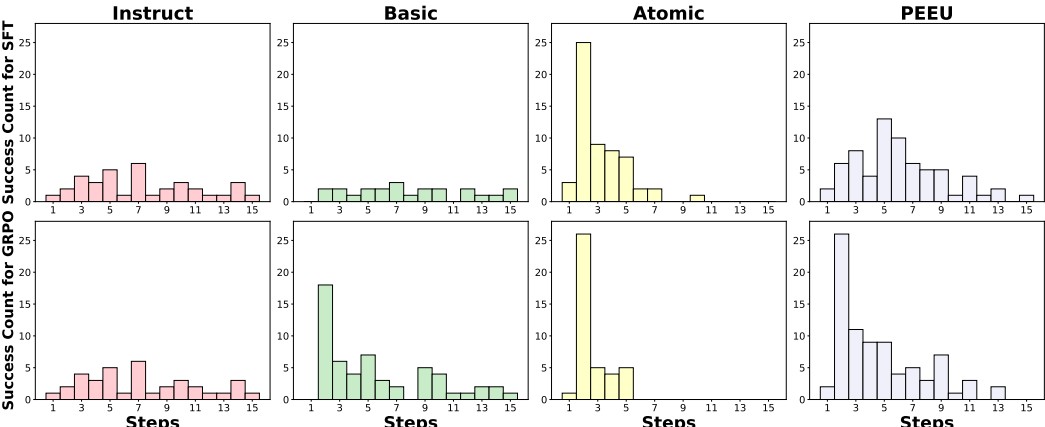

Figure 5: The distribution on the number of successful planning steps for 7B SFT and GRPO.

## 5 RELATED WORK

**DeepResearch Agent.** DeepResearch emphasizes broad web searches (Zhang et al., 2025b; Li et al., 2025c). Systems like WebSailor (Li et al., 2025a), WebShaper (Tao et al., 2025), and Web-Watcher (Geng et al., 2025) focus on information seeking. But experience summarization and compositional generalization analysis (Li et al., 2025b) remain underexplored. Agent KB (Tang et al., 2025b) and Memento (Zhou et al., 2025a) construct structured knowledge bases from past explorations using prompt engineering. We study compositional generalization in task planning and leverage experiences to train agents, enabling them to achieve stronger web-based planning capabilities under the same scale of data.

**Multimodal Web Navigation Agent.** The research on multimodal web agent navigation emphasizes vertical depth navigation on web pages (Wang et al., 2024; Ning et al., 2025; Tang et al., 2025a). Open-source models need two core abilities: grounding and planning. Some works strengthen grounding for more accurate spatial coordinates (Lu et al., 2025; Luo et al., 2025; Zhou et al., 2025b). The SoM representation can reduce the influence of grounding, making it easier to study improvements in planning ability. Prior work often trains on low-level tasks (Gu et al., 2024; Fan et al., 2025) or distills teacher trajectories without fully utilizing experiences (Logeswaran et al., 2025; Trabucco et al., 2025). Our approach makes high-level tasks more aligned and constrained, thereby providing stronger generalization ability.

## 6 CONCLUSION

In this work, we analyze the compositional generalization of MLLMs in web navigation planning tasks. Through the proposed **TDHAF** framework, it shows that high-level task training is essential for OOD generalization. Based on these findings, we introduce the **PEEU** method, which enables autonomous exploration and effective experience utilization. Experiments on real-world websites demonstrate that PEEU consistently outperforms baselines across model scales and training paradigms, highlighting the importance of leveraging high-level tasks to enhance planning ability.

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

## A    THE USE OF LARGE LANGUAGE MODELS

In this paper, we use ChatGPT to polish our writing and check for grammar errors. The authors are responsible for the contents of this submissions.

## B    DEFINITION DETAILS

In this section, we introduce and formalize the definitions of task planning, and then present the three levels of task planning granularity in this work, including low-level tasks, mid-level tasks, and high-level tasks. As well as the definitions of in-domain, out-of-domain and experience.

**Task Planning Definition.**    The task planning is formally defined as a tuple (Li et al., 2025d; Cao et al., 2025; Wei et al., 2025):

$$\mathcal{P} = \langle \mathcal{S}, \mathcal{A}, T, s_0, \mathcal{G} \rangle. \tag{7}$$

Here, $\mathcal{S}$ is a set of environment states, $\mathcal{A}$ is a set of actions, $T : \mathcal{S} \times \mathcal{A} \to \mathcal{S}$ is a state transition function, $s_0 \in \mathcal{S}$ is an initial state, $\mathcal{G} \subseteq \mathcal{S}$ is a set of goal states. The objective is to find a sequence of actions $\langle a_0, a_1, \ldots, a_n \rangle$ that transforms the system from the initial state $s_0$ to a goal state $s_g \in \mathcal{G}$.

In the ReAct paradigm (Yao et al., 2023), the objective is to output the next action given the task description, history, and current observation. This can be formally represented as:

$$a_t = \pi(d, \mathcal{H}_{0:t}, s_t). \tag{8}$$

Here, $d$ is the task description, and $\mathcal{H}_{0:t} = \{(s_0, a_0), (s_1, a_1), \ldots, (s_{t-1}, a_{t-1})\}$ is the history of state-action pairs up to time $t - 1$, $s_t$ is the current observation, and $\pi$ is the planning policy that outputs the action $a_t$. Upon task completion, we obtain a trajectory $\tau = \{(s_0, a_0), (s_1, a_1), \ldots, (s_n, a_n)\}$.

**Low-level Task Definition.**    The low-level task is defined as a single-step task. It is also called the atomic-level task. For step $t$, the policy $\pi$ uses only the current low-level task description $d_{low}$ and the current observation $s_t$ to determine the next action:

$$a_t = \pi(d_{low}, s_t). \tag{9}$$

**Mid-level Task Definition.**    The mid-level task is defined as a multi-step subtask. For a subtask spanning steps $p$ to $q$, the policy $\pi$ uses the middle-level task description $d_{mid}$, the history $\mathcal{H}_{p:t}$ and the current observation $s_t$ to determine the next action:

$$a_t = \pi(d_{mid}, \mathcal{H}_{p:t}, s_t). \tag{10}$$

**High-level Task Definition.**    The high-level task is defined as a long horizon, composed of a sequence of subtasks. For a long horizon task 0 to $n$, the policy $\pi$ uses the middle-level task description $d_{high}$, the history $\mathcal{H}_{0:t}$ and the current observation $s_t$ to determine the next action:

$$a_t = \pi(d_{high}, \mathcal{H}_{0:t}, s_t). \tag{11}$$

**In-Domain and Out-of-Domain.**    For the TDHAF, ID evaluation uses test data from the same trajectories seen during post-training. The task description has been paraphrased, while OOD evaluation uses test data from entirely new websites not encountered during post-training. For the PEEU, ID evaluation uses test data from the same websites seen during post-training, while OOD evaluation uses test data from entirely new websites not encountered during post-training.

**Experience Definition.**    As defined in Silver & Sutton (2025), experience is defined as data produced through an agent's interactions with the environment. Subsequent work (Cai et al., 2025) further categorizes experiences into trajectories, knowledge and skills summarized from these trajectories. In this paper, we mainly refer to what is summarized from the trajectory as experience.

## C  TDHAF DATASET DETAILS

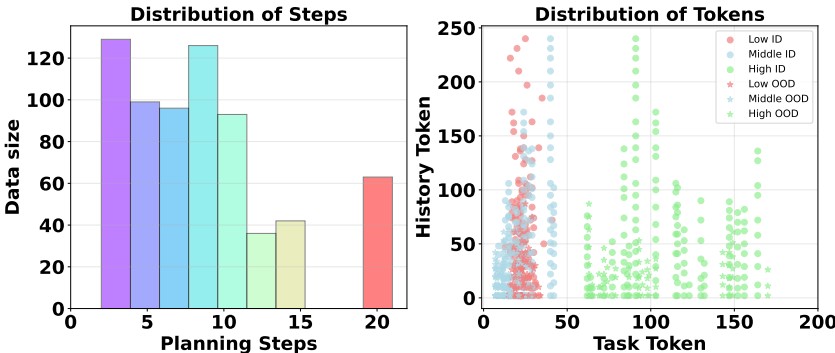

Figure 6: Data Distribution for TDHAF.

Table 3: This table shows the TDHAF division of training and test sets for three generalization dimensions. ID indicates that training and test are derived from the same trajectory in the same websites, but the tasks are rewritten. OOD indicates they come from different trajectories across different websites. L denotes low-level tasks, M denotes mid-level tasks, H denotes high-level tasks.

| Training Set | Test Set |
|---|---|
| **ID Bottom-up Generalization** | |
| Train-ID-L | Test-ID-L, Test-ID-M, Test-ID-H |
| Train-ID-M | Test-ID-M, Test-ID-H |
| Train-ID-H | Test-ID-H |
| **ID Top-down Generalization** | |
| Train-ID-L | Test-ID-L |
| Train-ID-M | Test-ID-L, Test-ID-M |
| Train-ID-H | Test-ID-L, Test-ID-M, Test-ID-H |
| **OOD Multi-level Generalization** | |
| Train-ID-L | Test-OOD-L, Test-OOD-M, Test-OOD-H |
| Train-ID-M | Test-OOD-L, Test-OOD-M, Test-OOD-H |
| Train-ID-H | Test-OOD-L, Test-OOD-M, Test-OOD-H |

## D  TDHAF PROMPT

**Build Low Level Prompt for TDHAF**

```
Your task is to generate task descriptions for CLICK/TYPE/SELECT an on-
    screen element.

Two screenshots are provided:
Current UI - Shows a interactive element (labeled "1") with a bounding
    box.
Post-interaction UI - Highlights changes after interaction (excluding
    bounding box disappearance).

Task:
Purpose Clarity - Clearly define the purpose of the interaction with
    the UI element in both descriptions, ensuring they are functionally
     identical but phrased differently.
```

```
Ensure the two descriptions serve distinct contexts with no overlapping
    phrasing.
Action Consistency - Use only CLICK, TYPE, or SELECT as action types,
    with identical parameters in both descriptions (e.g., target
    element, input text, or selection option).
UI Change Focus - Describe only observable UI changes (e.g., new
    elements appearing, data updates, transitions) resulting from the
    action-avoid vague or future-oriented statements.
Training vs. Testing Wording - Paraphrase the purpose distinctly for
    training (instructional) and testing (validation) contexts while
    keeping functional outcomes identical.
Now, generate the two mission-style descriptions adhering to these
    rules. Only output the lists, nothing else.

The raw task is <task>.
```

**Build High Level Prompt for TDHAF**

```
Please make this task more complex, but do not change the parameters in
    this task. Add more subtasks after this task, and rephrase the
    original task with synonymous expressions. This task and subsequent
    tasks can be combined into a more complex task. More complex means
    that the current task is a subtask in the middle, and then more
    subtasks are added before and after to merge into a more complex
    task. But don't describe the specific tasks in detail. Please
    output two task descriptions that are paraphrases of each other, in
    the form of a list of json. The key of the element is the string
    task, and the value is the task description.
The raw task is <task>.
```

**Inference Prompt for Multimodal-Mind2web for Agent**

```
User:
<image>You are a web agent.
Your task is: <task>
The history is: <history>.
If you want to complete the task, you should output action CLICK/TYPE/
    SELECT, id and value in <answer> </answer> tags. Output the one
    bbox you should interact with in JSON format.

Examples:
1. For clicking: <answer>{"action": "CLICK", "value": "" ,"id": 3}</
    answer>
2. For typing text: <answer>{"action": "TYPE","value": "example@email.
    com", "id": 5}</answer>
3. For selecting an option: <answer>{"action": "SELECT", "value": "
    United States","id": 2}</answer>
```

## E  GENERALIZATION DISTRIBUTION AND DEFINITION

Let the set of levels be

$$L = \{\text{low}, \text{middle}, \text{high}\}. \tag{12}$$

For a sample $x$ at level $\ell \in L$, define an indicator

$$I(\ell, x) = \begin{cases} 1, & \text{if the prediction at level } \ell \text{ is correct,} \\ 0, & \text{otherwise.} \end{cases} \tag{13}$$

**Good Generalization.**  The model is considered to generalize well at some level (low, middle, or high) if it predicts correctly not only at this level but also at the other two levels. That means correct

at all three levels. Good generalization means successful generalization to other levels, the larger the better.

$$\text{Good}(\ell, x) = 1 \quad \text{if and only if} \quad I(\ell', x) = 1 \ \forall \ell' \in L. \tag{14}$$

**Bad Generalization.** The model is considered to generalize bad at some level if it is correct at this level, but at least one of the other two levels is wrong. Bad generalization means failure to fully generalize to other levels, the smaller the better.

$$\text{Bad}(\ell, x) = 1 \quad \text{if and only if} \quad I(\ell, x) = 1 \text{ and } \exists \ell' \in L, \ \ell' \neq \ell \ \text{ with } I(\ell', x) = 0. \tag{15}$$

**Coverage Percentage.** Among all samples that are predicted correctly at their own level, and these samples that are also correct at all three levels (i.e., that achieve good generalization) is called the *coverage percentage*.

Formally, let

$$G_\ell = \{x \in S_\ell \mid \text{Good}(\ell, x) = 1\} \tag{16}$$

be the set of samples that are correctly predicted at level $\ell$ and also satisfy the good generalization condition. Here, $S_\ell$ denotes the set of all samples that are predicted correctly at level $\ell$, and $T$ denotes the entire test set.

The coverage percentage at level $\ell$ is then defined as

$$\text{Coverage}(\ell) = \frac{|G_\ell|}{|T|} \times 100\%. \tag{17}$$

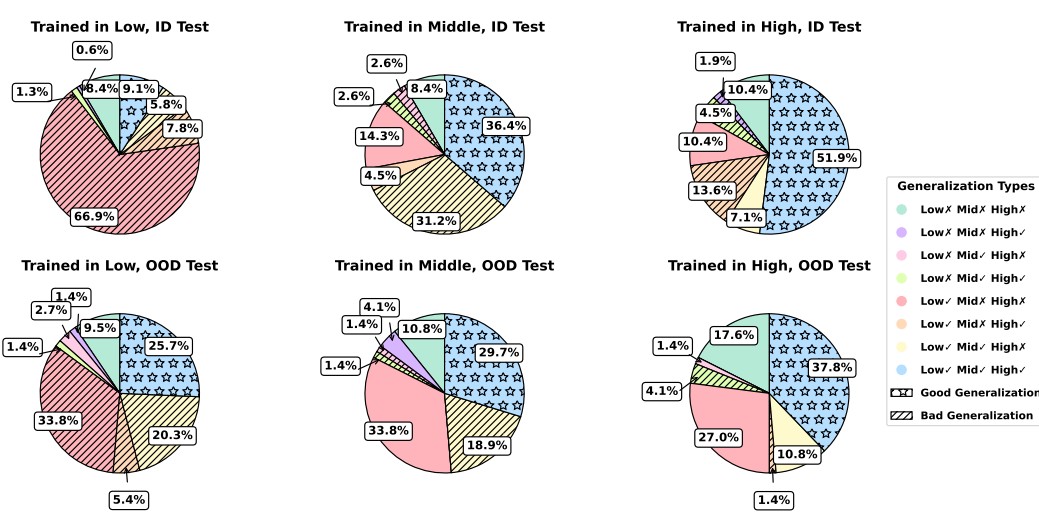

Figure 7: Generalization Distribution Pie Chart for Qwen2.5-VL-7B. Good generalization means successful generalization to other levels, and the larger it is, the better. Bad generalization means failure to fully generalize to other levels, and the smaller it is, the better.

## F PEEU PROMPT

**Inference Prompt for WebVoyager**

```
System:
Imagine you are a robot browsing the web, just like humans. Now you
    need to complete a task. In each iteration, you will receive an
    Observation that includes a screenshot of a webpage and some texts.
     This screenshot will feature Numerical Labels placed in the TOP
    LEFT corner of each Web Element.
```

```
Carefully analyze the visual information to identify the Numerical
    Label corresponding to the Web Element that requires interaction,
    then follow the guidelines and choose one of the following actions:
1. Click a Web Element.
2. Delete existing content in a textbox and then type content.
3. Scroll up or down. Multiple scrolls are allowed to browse the
    webpage. Pay attention!! The default scroll is the whole window. If
     the scroll widget is located in a certain area of the webpage,
    then you have to specify a Web Element in that area. I would hover
    the mouse there and then scroll.
4. Wait. Typically used to wait for unfinished webpage processes, with
     a duration of 5 seconds.
5. Go back, returning to the previous webpage.
6. Google, directly jump to the Google search page. When you can't find
     information in some websites, try starting over with Google.
7. Answer. This action should only be chosen when all questions in the
    task have been solved.

Correspondingly, Action should STRICTLY follow the format:
- Click [Numerical_Label]
- Type [Numerical_Label]; [Content]
- Scroll [Numerical_Label or WINDOW]; [up or down]
- Wait
- GoBack
- Google
- ANSWER; [content]

Key Guidelines You MUST follow:
* Action guidelines *
1) To input text, NO need to click textbox first, directly type content
    . After typing, the system automatically hits `ENTER` key.
    Sometimes you should click the search button to apply search
    filters. Try to use simple language when searching.
2) You must Distinguish between textbox and search button, don't type
    content into the button! If no textbox is found, you may need to
    click the search button first before the textbox is displayed.
3) Execute only one action per iteration.
4) STRICTLY Avoid repeating the same action if the webpage remains
    unchanged. You may have selected the wrong web element or numerical
     label. Continuous use of the Wait is also NOT allowed.
5) When a complex Task involves multiple questions or steps, select "
    ANSWER" only at the very end, after addressing all of these
    questions (steps). Flexibly combine your own abilities with the
    information in the web page. Double check the formatting
    requirements in the task when ANSWER.
* Web Browsing Guidelines *
1) Don't interact with useless web elements like Login, Sign-in,
    donation that appear in Webpages. Pay attention to Key Web Elements
     like search textbox and menu.
2) Vsit video websites like YouTube is allowed BUT you can't play
    videos. Clicking to download PDF is allowed and will be analyzed by
     the Assistant API.
3) Focus on the numerical labels in the TOP LEFT corner of each
    rectangle (element). Ensure you don't mix them up with other
    numbers (e.g. Calendar) on the page.
4) Focus on the date in task, you must look for results that match the
    date. It may be necessary to find the correct year, month and day
    at calendar.
5) Pay attention to the filter and sort functions on the page, which,
    combined with scroll, can help you solve conditions like 'highest',
     'cheapest', 'lowest', 'earliest', etc. Try your best to find the
    answer that best fits the task.

For example:
```

```
Click [3]
Type [3]; [apple]
Scroll [WINDOW]; [down]
Wait
GoBack
Google
ANSWER; [apple is red]

Your reply should strictly follow the format:
Thought: {Your brief thoughts (briefly summarize the info that will
    help ANSWER)}
Action: {One Action format you choose}

Then the User will provide:
Observation: {A labeled screenshot Given by User}

User:
<image>Now given a task: <task> Please interact with https://www.
    example.com and get the answer. Observation: please analyze the
    attached screenshot and give the Thought and Action. I've provided
    the tag name of each element and the text it contains (if text
    exists). Note that <textarea> or <input> may be textbox, but not
    exactly. Please focus more on the screenshot and then refer to the
    textual information. <SoM Observation>
```

**Task Setting Prompt**

```
<image>
Analyze the given webpage screenshot and generate 50 different tasks
    that users might want to accomplish on this website.
You can focus on searching for specific items. The task should be
    combined with the specific function of this website.
The tasks should be varied, and there should be both difficult and
    simple tasks.
Output only a JSON-formatted list of tasks with no additional
    commentary or explanation.
Example format:
{
"tasks": [
"task 1 description",
"task 2 description",
...
"task n description"
]
}
```

**Exploration Prompt**

```
Imagine you are a robot browsing the web, just like humans. Now you
    need to complete a task. In each iteration, you will receive an
    Observation that includes a screenshot of a webpage and some texts.
     This screenshot will feature Numerical Labels placed in the TOP
    LEFT corner of each Web Element.
Carefully analyze the visual information to identify the Numerical
    Label corresponding to the Web Element that requires interaction,
    then follow the guidelines and choose one of the following actions:
1. Click a Web Element.
2. Delete existing content in a textbox and then type content.
3. Scroll up or down. Multiple scrolls are allowed to browse the
    webpage. Pay attention!! The default scroll is the whole window. If
     the scroll widget is located in a certain area of the webpage,
```

```
        then you have to specify a Web Element in that area. I would hover
        the mouse there and then scroll.
4. Wait. Typically used to wait for unfinished webpage processes, with
   a duration of 5 seconds.
5. Go back, returning to the previous webpage. If you scroll down more
   than twice and still can't find the answer, you need to use "Go
   back" to return.
6. Google, directly jump to the Google search page. When you can't find
    information in some websites, try starting over with Google.
7. Answer. This action should only be chosen when all questions in the
   task have been solved.

Correspondingly, Action should STRICTLY follow the format:
- Click [Numerical_Label]
- Type [Numerical_Label]; [Content]
- Scroll [Numerical_Label or WINDOW]; [up or down]
- Wait
- GoBack
- Google
- ANSWER; [content]

Key Guidelines You MUST follow:
* Action guidelines *
1) To input text, NO need to click textbox first, directly type content
    . After typing, the system automatically hits 'ENTER' key.
   Sometimes you should click the search button to apply search
   filters. Try to use simple language when searching.
2) You must Distinguish between textbox and search button, don't type
   content into the button! If no textbox is found, you may need to
   click the search button first before the textbox is displayed.
3) Execute only one action per iteration.
4) STRICTLY Avoid repeating the same action if the webpage remains
   unchanged. You may have selected the wrong web element or numerical
    label. Continuous use of the Wait is also NOT allowed.
5) When a complex Task involves multiple questions or steps, select "
   ANSWER" only at the very end, after addressing all of these
   questions (steps). Flexibly combine your own abilities with the
   information in the web page. Double check the formatting
   requirements in the task when ANSWER.
6) If you feel the current product does not meet the task requirements,
    you can use GoBack action to return to the previous screen and
   look for other products. Don't just scroll down-learn to go back.
* Web Browsing Guidelines *
1) Don't interact with useless web elements like Login, Sign-in,
   donation that appear in Webpages. Pay attention to Key Web Elements
    like search textbox and menu.
2) Vsit video websites like YouTube is allowed BUT you can't play
   videos. Clicking to download PDF is allowed and will be analyzed by
    the Assistant API.
3) Focus on the numerical labels in the TOP LEFT corner of each
   rectangle (element). Ensure you don't mix them up with other
   numbers (e.g. Calendar) on the page.
4) Focus on the date in task, you must look for results that match the
   date. It may be necessary to find the correct year, month and day
   at calendar.
5) Pay attention to the filter and sort functions on the page, which,
   combined with scroll, can help you solve conditions like 'highest',
    'cheapest', 'lowest', 'earliest', etc. Try your best to find the
   answer that best fits the task.

Your reply should strictly follow the format:
Thought: {Your brief thoughts (briefly summarize the info that will
    help ANSWER)}
Action: {One Action format you choose}
```

```
Then the User will provide:
Observation: {A labeled screenshot Given by User}
```

**Experience Summarize Prompt**

```
Analyze the user's intent based on the following:
The action performed between these interfaces is <ACTION>

Task:
The first screenshot shows the interface before interaction, while the
    second screenshot displays the interface after the click operation.
Generate descriptions explaining the purpose of interaction with the
    element.
Focus on meaningful UI changes (e.g., new elements, transitions, or
    data updates, Don't pay attention to the changes in the bbox.).
Only output the task descriptions experience.
```

**Experience Utilization Prompt**

```
In this task, there are too many details provided. I only want to keep
    the details specified by the user, and the specific operational
    details need to be deleted.
Please directly output the processed string.
The task requirement is a declarative sentence, appearing like a real
    world user task.

The raw task is as follows:<low-level task list>
```

# G  ALGORITHM DETAILS

In this algorithm, the number of tasks is set to 100, and the maximum exploration depth is 15. In the experiments, exploration is performed on one website, while testing is conducted on 12 previously unseen websites to evaluate cross-site generalization ability. The algorithm is shown in Algorithm 1.

---

**Algorithm 1** Autonomous Planning with Exploration and Experience Utilization

---

**Require:** Website URL, MLLM $M$, Environment Env
**Ensure:** Policy $\pi$ for task-oriented planning
    **Stage 1: Planning Tree Exploration**
 1: Obtain homepage state $s_0$ from the given URL
 2: Generate task list: $\mathcal{D} = M(s_0, \text{URL})$
 3: **for** each task $d_i \in \mathcal{D}$ **do**
 4:    Execute actions $a_t$ guided by $M$
 5:    Transition: $s_{t+1} \sim P(\cdot|s_t, a_t)$
 6:    Record trajectory $\tau = (s_0, a_0, s_1, a_1, \dots)$
 7: **end for**
 8: Build exploration tree $\mathcal{R} = \text{Explore}(M, \mathcal{T}, \text{Env}, \text{URL})$
    **Stage 2: Planning Experience Utilization**
 9: **for** each trajectory $\tau$ **do**
10:    Extract atomic experiences $\epsilon_t = (s_t, a_t, s_{t+1})$
11:    Build $\mu = (\epsilon_0, \epsilon_1, \dots, \epsilon_T)$
12:    Fuse into PEEU task: $\tilde{d} = \Phi(\mu)$
13: **end for**
14: Train policy $\pi$ with SFT and GRPO using PEEU dataset
15: **return** trained policy $\pi$

---

For RL training, we set two types of rewards. The first reward is for format, and the second reward is for answer correctness. For the format reward, we align with the action space and action format from WebVoyager. Each reward is 1.0, and if both are correct, the total reward is 2.0.

$$r_{\text{format}} = \begin{cases} 1.0, & \text{if the action follows the predefined format} \\ 0.0, & \text{otherwise,} \end{cases} \tag{18}$$

$$r_{\text{answer}} = \begin{cases} 1.0, & \text{if the predicted answer is correct} \\ 0.0, & \text{otherwise,} \end{cases} \tag{19}$$

$$R_{rl} = r_{\text{format}} + r_{\text{answer}}. \tag{20}$$

## H  PEEU EXPERIMENT DETAILS

We evaluate the planning capabilities of the models on real-world multimodal benchmark WebVoyager (He et al., 2024). The test set consists of 643 samples, covering 15 real multimodal online websites, including shopping, research, code, travel, and other categories. Because some websites have access frequency limits, we do not evaluate Cambridge Dictionary and Google Search. We train only on Allrecipes and test on this site and the remaining 12 websites. Following WebVoyager (He et al., 2024), these websites fully comply with the terms of service and user agreements. The exploration task includes 100 items with a maximum of 15 steps. We filter out data with wrong formats. The final training set size is 579. In this section, ID refers to the same websites but different tasks, while OOD refers to completely unseen and different websites. The abbreviation corresponds to the following names, as shown in Table 4.

| Abbreviation | Full Name | Domain | Category |
|---|---|---|---|
| Rec | Allrecipes | ID | Cooking |
| Ama | Amazon | OOD | Shopping |
| App | Apple | OOD | Shopping |
| ArX | ArXiv | OOD | Research |
| Git | GitHub | OOD | Code |
| Boo | Booking | OOD | Travel |
| ESP | ESPN | OOD | Sports |
| Cou | Coursera | OOD | Study |
| BBC | BBC News | OOD | News |
| Fli | Google Flights | OOD | Travel |
| Map | Google Map | OOD | Map |
| Hug | Huggingface | OOD | Model |
| Wol | Wolfram | OOD | Tool |
| Overall | Average accuracy of all websites | ID/OOD | Diversity |

Table 4: Abbreviations and corresponding full names table

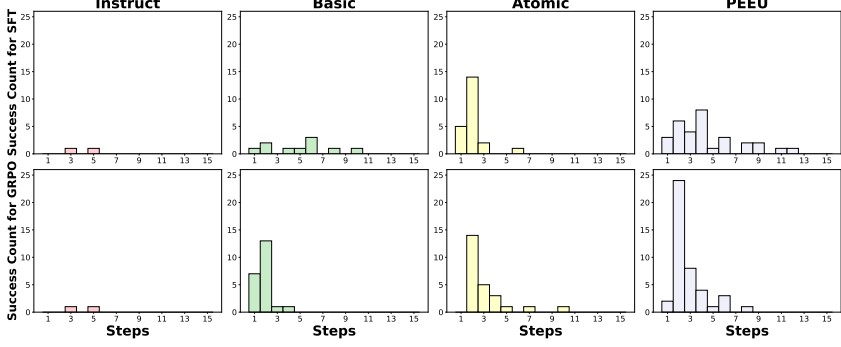

Figure 8: The distribution on the number of successful planning steps for 3B SFT and GRPO.

# I  TRAINING REWARD DETAILS

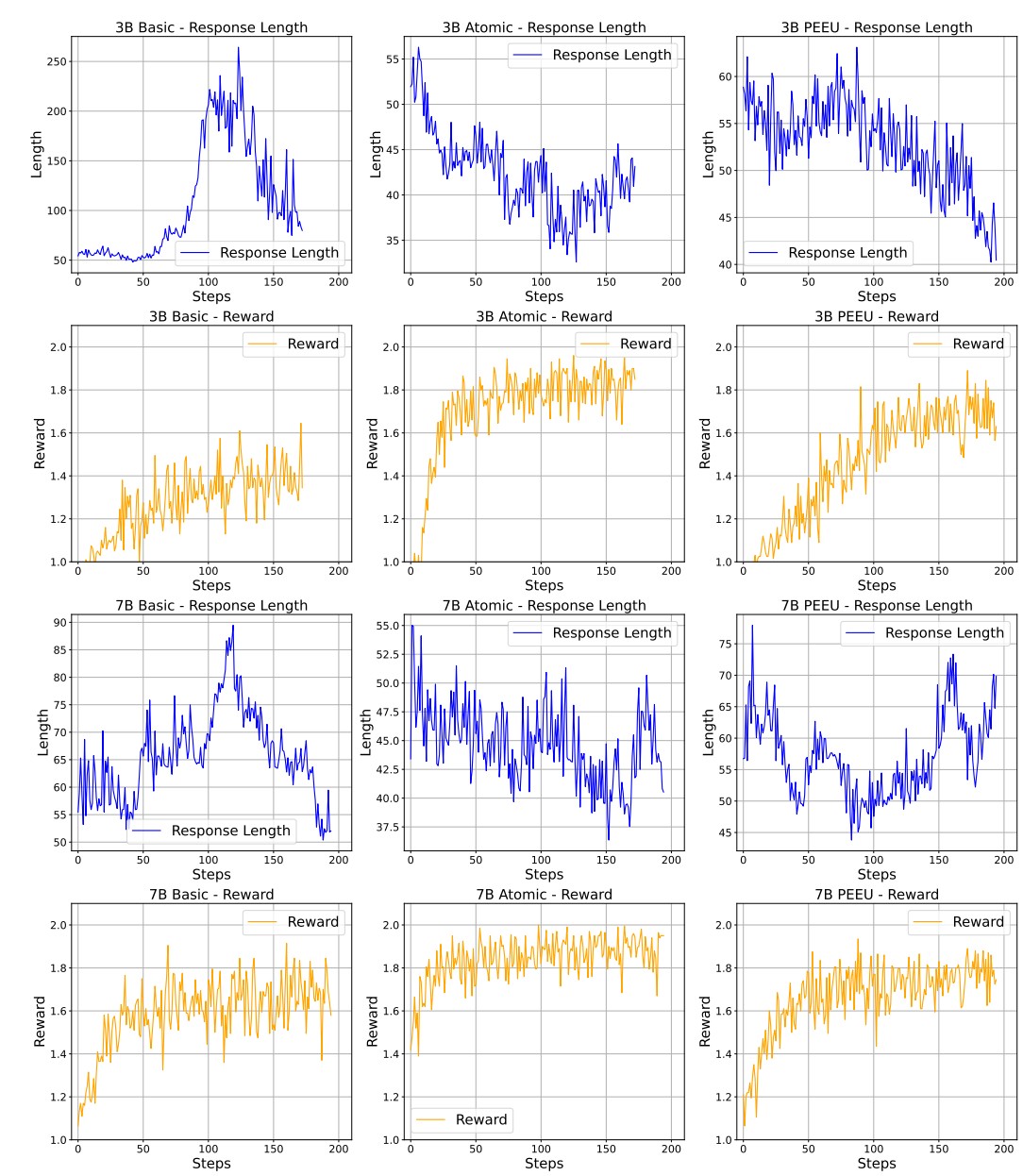

Figure 9: RL Training Reward.

