# OpenReview forum: "WebPlanner: Task Planning with Autonomous Experience Exploration and Utilization for Real World Multimodal Web Agents"
_ICLR.cc/2026/Conference — ICLR 2026 Conference Withdrawn Submission_

### Official Review · Reviewer_KhVw · 2025-10-30

**Soundness:** 1
**Presentation:** 1
**Contribution:** 1
**Rating:** 2
**Confidence:** 4

**Summary:**

This work investigates how to improve the generalizability of small models on web tasks, motivated by the idea that these models should learn tasks at different granularities. The authors find that learning high-granularity (task-level) objectives can improve out-of-domain (OOD) generalization, and therefore propose that small models should learn to perform planning in web-based environments. The paper then tests this idea experimentally.

**Strengths:**

I think the problem explored — enabling small models used for web tasks to generalize to unseen environments — is an important one. Moreover, the motivation of training models on high-level, task-granular data (i.e., performing task-level planning) is reasonable and conceptually sound.

**Weaknesses:**

First, the experiments are limited and the results are weak. The work feels unfinished — as if the experiments were stopped halfway and the paper was quickly wrapped up and submitted. Notably, the best result reported in the abstract (14.9%) is less than one-third of that achieved by closed-source models, and the improvement over the only included baseline is not substantial.

Several relevant baselines are missing, such as memory-based agents (e.g., AWM) and trainable multimodal web agents trained on their own curated datasets. Because of these omissions, the experimental tables supporting the main claims are too thin to convincingly sustain the paper’s conclusions.

Second, the core idea is not novel. Many existing agent frameworks have already proposed decoupled learning of high-level planning and low-level reasoning, including trainable multimodal web agents that separately train these two modules. The proposed method for improving small-model generalization is essentially another instantiation of this mainstream paradigm and does not differ meaningfully from prior approaches.

Finally, the writing quality could be improved. The most crucial figure explaining the experimental idea is confusing and poorly organized, while the motivation section is much more complete than the sections describing the main results and conclusions.

**Questions:**

Are there additional baseline results not reported in the paper?

How exactly does this approach relate to prior works that use stepwise, independent learning of high-level planning and low-level reasoning? This should be clarified and strengthened in the related work section.

---

### Official Review · Reviewer_3SeQ · 2025-11-01

**Soundness:** 3
**Presentation:** 3
**Contribution:** 2
**Rating:** 4
**Confidence:** 4

**Summary:**

The paper introduces a multi-level evaluation framework to study in-domain and out-of-domain generalization, and presents Planning Experience Exploration and Utilization (PEEU), an automatic method for collecting and leveraging planning knowledge from real web interactions.

**Strengths:**

1. The paper is well-written and clearly presented, with high-quality figures and visualizations that effectively illustrate the framework and results.

2. It provides a systematic analysis of model generalization, proposing three task levels and studying both in-domain and out-of-domain generalization. The results offer valuable insights for understanding the compositional and hierarchical generalization ability of multimodal planners.

**Weaknesses:**

1. While the overall presentation of the paper is decent, several figures lack clarity. For instance, the legend in Figure 1a is unclear, Figures 3 and 7 are difficult to understand, and the dashed lines in Figures 1 and 2 are not well explained.

2. The experimental setup in Section 3 appears questionable. Training for only three epochs may not be sufficient for convergence, and using Step SR as a main metric seems too coarse and may not reflect true model performance for modern large models.

3. The paper’s overall structure lacks clear logical flow. The proposed TDHAF framework seems somewhat disconnected from the later PEEU design, and the relationship between the two parts is not well justified.

4. The experiments are not fully convincing. The models used (Qwen 2.5 3B/7B) are relatively small for web-based reasoning tasks, and the proprietary models included are no longer state-of-the-art. Including results from stronger closed models such as Claude 4 or GPT-5, and at least larger open models (e.g., 32B level), would make the comparison more meaningful.

5. The overall performance is not very promising, as it does not outperform other recent frameworks [1] that also adopt self-training or task generation independent of specific benchmarks.

[1] Proposer-Agent-Evaluator(PAE): Autonomous Skill Discovery For Foundation Model Internet Agents. ICML 2025.

**Questions:**

1. How does the tree exploration module handle the case where performing the same action on the same page may lead to different results at different times (e.g., due to dynamic web content or time-sensitive updates)?

2. How do the authors ensure the quality of tasks generated by PEEU? Are there any quantitative measures of task validity or similarity to human-curated benchmarks?

3. Could the authors provide a clearer explanation of the figures, especially Figure 1a, 3, and 7? What do the dashed lines represent?

---

### Official Review · Reviewer_vJoe · 2025-11-01

**Soundness:** 2
**Presentation:** 3
**Contribution:** 2
**Rating:** 4
**Confidence:** 3

**Summary:**

This paper proposes WebPlanner, a framework comprising the TDHAF dataset for hierarchical web tasks and the PEEU method for autonomous exploration, aiming to enhance out-of-distribution (OOD) task planning and generalization capabilities of small-scale multimodal large language models (MLLMs).

**Strengths:**

1.Well-structured task hierarchy: TDHAF clearly defines three levels of task difficulty (low, medium, high) and distinguishes between in-distribution (ID) and out-of-distribution (OOD) evaluation settings, providing a structured benchmark for assessing compositional generalization.

2.Focus on practical, small-scale models: The work targets efficient MLLMs (3B–7B parameters), aligning with real-world deployment needs rather than relying solely on massive models.

3.Promising autonomous exploration concept: PEEU introduces the idea of “given a URL, the agent can freely explore and set sub-goals,” reducing reliance on fully supervised trajectories—a potentially valuable direction.

4.Comprehensive generalization evaluation: The experiments cover bottom-up, top-down, and multi-level generalization scenarios, offering nuanced insights into how training strategies affect transfer across abstraction levels.

**Weaknesses:**

Severely under-specified PEEU mechanism:
1.Exploration triggering is unclear: When does the model decide to “explore” vs. “execute”? Is it based on uncertainty, task failure, or heuristic rules?

2.No reward design described: The paper omits any discussion of reward signals—critical for guiding exploration or evaluating task completion. Without this, it’s unclear whether PEEU is truly self-supervised or implicitly relies on human-labeled trajectories.

3.Exploration-exploitation trade-off is ignored: How does the agent avoid unproductive clicking or infinite loops? Are there exploration budgets or termination conditions?

The work reads more like a system pipeline (dataset + training protocol) than a principled algorithmic advance. It lacks analysis of why hierarchical task decomposition improves OOD generalization or connections to established theories in reinforcement learning or curriculum learning.

**Questions:**

1.How are sub-goals generated in PEEU? Are they produced by the model itself, or derived from predefined templates? Please clarify the mechanism.

2.Is there a reward signal during exploration? If yes, what is the exact form of the reward function? If not, how does the agent learn to move toward task completion?

3.During the PEEU phase, does the model interact with live websites (online exploration), or operate only on static HTML snapshots or simulated environments?

4.Do the “high-level” tasks in TDHAF truly require multi-hop reasoning? For example, does “book a highly rated room” involve genuine planning, or is it merely a sequential execution of “check rating → select room → book”?

---

### Official Review · Reviewer_d4P7 · 2025-11-02

**Soundness:** 3
**Presentation:** 2
**Contribution:** 2
**Rating:** 4
**Confidence:** 3

**Summary:**

This paper investigates generalization capabilities of small-scale multimodal large language models for web navigation tasks. The authors propose (1) a Task Decomposition Hierarchical Analysis Framework (TDHAF) to evaluate compositional generalization across three task granularities (low-, mid-, high-level) and two generalization types (in-domain, out-of-domain), and (2) a Planning Experience Exploration and Utilization (PEEU) method, where a stronger model autonomously explores websites, extracts experience trajectories, and synthesizes high-level tasks to train smaller MLLMs, which shows improved performance on WebVoyager.

**Strengths:**

- Generalization is an important and interesting topic, and the paper provides a structured analysis of hierarchical generalization, covering ID Bottom-up Generalization, ID Top-down Generalization, and OOD Multi-level Generalization.

**Weaknesses:**

- The proposed “exploration” stage simply relies on a model to propose candidate tasks, collect trajectories, and refine the collected data, which is a very common practice in recent web navigation literature. Similar exploration-driven trajectory synthesis has already been explored in works such as Explorer [1].
Likewise, using the teacher model to summarize low-level steps into high-level task formulations is conceptually similar to many existing planning-based methods [2].
Compared to these prior methods, this paper does not introduce any novel exploration strategies.
---
References:
[1] Pahuja, Vardaan, et al. Explorer: Scaling exploration-driven web trajectory synthesis for multimodal web agents. arXiv:2502.11357 (2025).
[2] Erdogan, Lutfi Eren, et al. Plan-and-Act: Improving planning of agents for long-horizon tasks. arXiv:2503.09572 (2025).

**Questions:**

- When claiming that training on high-level versus low-level tasks improves generalization, have you explored other formulations of action representation? For example using click(x, y) instead of click(button_id), which would require the models to have stronger low-level grounding abilities.

---

### Note · Authors · 2025-12-28

I have read and agree with the venue's withdrawal policy on behalf of myself and my co-authors.